# Ashwin Gene Expression Profiles in Oocytes, Preimplantation Embryos, and Fetal and Adult Bovine Tissues

**DOI:** 10.3390/ani10020276

**Published:** 2020-02-11

**Authors:** Verónica Moreno-Brito, Daniel Morales-Adame, Elier Soto-Orduño, Susana Aideé González-Chávez, César Pacheco-Tena, Gerardo Pavel Espino-Solis, Irene Leal-Berumen, Everardo González-Rodríguez

**Affiliations:** 1Faculty of Medicine and Biomedical Sciences, Autonomous University of Chihuahua, Circuito Universitario, Campus II, Chihuahua C.P. 31109, Chih., Mexico; vmoreno@uach.mx (V.M.-B.); sagonzalez@uach.mx (S.A.G.-C.); dr.cesarpacheco@gmail.com (C.P.-T.); gespinos@uach.mx (G.P.E.-S.); ileal@uach.mx (I.L.-B.); 2Faculty of Zootechnics and Ecology, Autonomous University of Chihuahua, Francisco R. Almada Km 1, Chihuahua C.P. 31453, Chih., Mexico; danielmorales_27@hotmail.com; 3Faculty of Chemical Sciences, Autonomous University of Chihuahua, Circuito Universitario, Campus II, Chihuahua C.P. 31109, Chih., Mexico; elier_soto@hotmail.com; 4Translational Research Laboratory, National Laboratory of Flow Cytometry, Autonomous University of Chihuahua, Circuito Universitario, Campus II, Chihuahua C.P. 31109, Chih., Mexico

**Keywords:** Ashwin, bovine, tissue expression, early embryonic development

## Abstract

**Simple Summary:**

*Ashwin* is a gene involved in the morphogenesis of the central nervous system and the early embryonic development of *Xenopus laevis*. The analysis of its phylogeny in silico has shown that its functions are restricted to vertebrates, but we lack additional information regarding its biological importance in higher vertebrates, such as mammals. The present study reveals the wide and variable expression of this gene in different bovine organs and confirms its significant expression during early embryonic development, with a pattern similar to that reported for maternal genes. In addition, specific expression of this gene throughout follicular development and during bovine spermatogenesis is revealed, leading to the proposal of *ashwin* as a new gene with important biological implications in bovine development and reproduction.

**Abstract:**

The *ashwin* gene, originally identified in *Xenopus laevis*, was found to be expressed first in the neural plate and later in the embryonic brain, eyes, and spinal cord. Functional studies of *ashwin* suggest that it participates in cell survival and anteroposterior patterning. Furthermore, *ashwin* is expressed zygotically in this species, which suggests that it participates in embryonic development. Nevertheless, the expression of this gene has not been studied in mammals. Thus, the aim of this study was to analyze the *ashwin* expression pattern in bovine fetal and adult tissues, as well as in three independent samples of immature and mature oocytes, and in two- to four-, and eight-cell embryos, morula, and blastocysts. Spatiotemporal expression was analyzed using real-time polymerase chain reaction (PCR); *ashwin* mRNA was detected in all tissues analyzed, immature and mature oocytes, and two- to eight-cell embryos. It was down-regulated in morula and blastocysts, suggesting that this expression profile is similar to that of maternal genes. Immunohistochemical localization of the *ashwin* protein in fetal and adult ovaries and testes reveals that this protein is consistently present during all stages of follicular development and during bovine spermatogenesis. These observations lead us to propose *ashwin* as an important gene involved in mammalian reproduction.

## 1. Introduction

Dynamic changes in genetic expression play important regulatory roles during development and cellular determination. Thus, identification of the genes and signaling systems that regulate these cellular events is essential. Although significant advances have been made in understanding how these events are coordinated, gaps remain in our understanding of how differentiation is coordinated.

The *ashwin* gene was isolated by differential display of genes activated in the neural specification of *Xenopus laevis* [1]. This gene encodes a 226-amino-acid protein that contains no previously identified domain, but does contains a leucine-rich N-terminal region adjacent to a basic region, which is similar to that of basic leucine zipper-like transcription factors [1,2]. Transcripts were detected in a broad dorsolateral domain after the initiation of gastrulation, in the dorsal hemisphere by midgastrulation, and in the circumblastoral ring and tail bud, with persistence in the brain, spinal cord, and eye, thereafter [1]. Functional assays involving the induction of overexpression by mRNA microinjection or translation inhibition with a morpholino oligonucleotide have revealed developmental truncations, incomplete neural tube closure, complete loss of the head structures, eye defects, asymmetrical development of bilateral structures, and a reduction in embryo size [1]. Together, these results indicate that the expression of *ashwin* coincides with neuronal induction and previous posterior patterning [1].

Moreover, spatial and temporal monitoring of the transcripts encoding this protein revealed that the gene is expressed maternally in unfertilized eggs, with sustained expression throughout early development in *X. laevis*, suggesting a role in early embryonic development [1]. However, the molecular mechanisms by which this protein acts remain unknown. Certain lines of evidence suggest that the *ashwin* protein, alone or jointly with noggin, generates synergistic expression of the neural cell adhesion molecule (NCAM) [3], zic3 [4], and noggin [1,5]. Noggin is a pleiotropic factor expressed in mammals in early and later stages of development; it antagonizes the action of bone morphogenetic proteins (BMPs), particularly BMP-2, -4, and -7, leading to a dorsal-ventral BMP gradient with subsequent germ layer formation that is essential for neural tube, tooth, hair follicle, and eye development [6]. Additionally, BMPs have been shown to regulate mammalian follicular development by affecting granulosa cell proliferation and steroidogenesis [7]. In silico analyses of nucleotide sequences have not led to the identification of conserved *ashwin* sequences in the *Drosophila*, *Caenorhabditis elegans*, or *Ciona* genome, but orthologous sequences have been identified in mice, humans, and bovines [1], implying that the functions of this gene observed in *X. laevis* are restricted to vertebrates. This evidence also suggests that *ashwin* regulates the molecular mechanisms by which noggin and BMPs act in mammals, such as bovines.

However, we lack basic information about the importance of *ashwin* in these processes. In this context, the purposes of this study were to analyze temporal and spatial patterns of *ashwin* expression in bovine fetal and adult tissues by reverse-transcription quantitative polymerase chain reaction (RT-qPCR) and to observe differential expression of *ashwin* in mature oocytes and embryos fertilized in vitro. In addition, we examined the expression of this protein during the development of germ cells in the bovine ovary and testicle.

## 2. Materials and Methods

### 2.1. Organ Samples

Samples of ovary, testis, heart, spleen, lung, muscle, kidney, and brain were collected from fetal (at 6 gestational months) and adult bovines at a local slaughterhouse. The tissues were immediately frozen in liquid nitrogen and stored at –80 °C until total RNA extraction.

### 2.2. Total RNA Isolation and Polymerase Chain Reactions

Total RNA from tissue biopsy samples was isolated with Trizol reagent^®^ (Invitrogen, Carlsbad, CA, USA; Thermo Scientific, Waltham, MA, USA) according to the manufacturer’s guidelines. Total RNA was also obtained from three independent samples of immature and mature oocytes over 24 h, and from two- to four-, and eight-cell embryos, morulas, and blastocysts, using the Arcturus Picopure extraction kit (Applied Biosystems, Foster City, CA, USA). Total RNA from tissue (1.0 μg) was used to synthesize cDNA. The cDNA (2 uL) was then used with specific primers to amplify *ashwin* (forward, 5’-ATGGCGGGGGATGTGGGCGG-3′; reverse, 5′-TCACGGCCATGTCACATGCT-3′) and β-actin (forward, 5′-GTGTGACATCAAGGAGAAGC-3′; reverse, 5′-TGGAAGGTGGACAGGGAGGC-3’). The amplification conditions were: initial denaturation at 90 °C for 1 min, followed by 35 cycles of denaturation at 95 °C for 1 min, annealing at 54 °C (*ashwin*) or 45 °C (β-actin) for 1 min, and extension at 60 °C for 1 min.

RT-qPCR was performed and reported according to the MIQE guidelines (Minimum Information for Publication of Quantitative Real-Time PCR Experiments). Following extraction of total RNA, a bioanalyzer (Agilent 2100; Santa Clara, CA, USA) was used to obtain an RNA integrity number (RIN) for each sample. All samples with RIN values >5.5 were retained for further processing. RNA yields were determined using a NanoDrop 2000 spectrophotometer (Thermo Fisher Scientific, Wilmington, DE, USA). cDNA was synthesized from 10 ng RNA in a 20 μL reaction volume with Ambion^®^ ArrayScript™ reverse transcriptase (10,000 units), random decamers (50 µM; both from Ambion Life Technologies), and dNTP mix (2.5 mM; Invitrogen) at 42 °C for 2 h. The enzyme was inactivated by incubation at 95 °C for 5 min. All samples were diluted at a 1:25 ratio with 1× TE buffer (Tris-EDTA) (pH 8.0) (Thermo Fisher Scientific, Wilmington, DE, USA). RT-PCR reactions were performed in 96-well plates with a final volume of 10 μL per well, using 2 μL cDNA diluted 1:25 and Gene Expression Master Mix (Applied Biosystems^®^). The master mix and probe concentrations were used according to the manual provided with the reagent kit. RT-qPCRs were performed with a real-time instrument (ABI PRISM 7900 HT; Applied Biosystems) using the following parameters: 1 cycle at 50 °C for 2 min, holding at 95 °C for 10 min, and 40 cycles at 95 °C for 5 s with annealing/extension at 60 °C for 1 min. The β-actin gene was used as an internal control to normalize quantification. The data are expressed as gene expression, where ΔCt = (Ct target – Ct reference), ΔΔCt = (ΔCt sample – ΔCt control), and gene expression = 2^-ΔΔCt^. 

### 2.3. Oocyte Maturation, In Vitro Fertilization, and Embryo Culture

Three independent collections of bovine cumulus-oocyte complexes (COCs) were retrieved by dissection of 70–80 ovaries collected from Angus cattle at a local slaughterhouse. Approximately 500–600 COCs with at least three layers of cumulus cells were aspired from 2 mm and 8 mm antral follicles using a PrecisionGlide needle (18 × 11/2”; Becton Dickinson) with vacuum suction (50 mmHg pressure; standard-duty WOB-L^®^ dry vacuum pumps; Welch^®^). Groups of 50 COCs were cultured in four-well plates (Nunc; Thermo Scientific, Rockford, IL, USA) and matured in 1 mL of chemically defined tissue culture medium (CDM, Chemically Defined Medium) supplemented with 5% fatty-acid-free bovine serum albumin (A7030 FAF-BSA; Sigma, St. Louis, MO, USA), follicle-stimulating hormone (15 ng/mL; Sioux Biochemical, Sioux Center, IA, USA), estradiol-17β (E2257, 0.1 μg/μL; Sigma), long-epidermal growth factor (E-9644; 50 ng/μL; Sigma), 2 mM D-fructose, 2.77 mM myo-inositol, 0.1 mM taurine, and cystamine (E-2257, 0.1 mM; Sigma)). The COCs were then incubated at 39 °C with 5% CO_2_ for 23 h.

Three rounds of fertilization were performed from three independent COC isolates. Frozen semen from an Angus bull was thawed and separated on a Percoll gradient in SpermTALP medium (P-1644; Sigma), then resuspended in CDM for in vitro fertilization (CDM supplemented with 0.5 mM D-fructose, 2 mM caffeine, 5% BSA (Bovine Serum Albumin), and 2 μg/mL (14 mM) NaCl) at a final concentration of 1 × 10^6^ sperm/mL. For fertilization, groups of 50 mature oocytes in four-well plates were co-incubated in microdrops (50 μL) of the sperm suspension for 18 h at 38.5 °C with 5% CO_2_. Presumptive zygotes were transferred to 0.5 mL microcentrifuge tubes containing 100 μL HEPES (4-(2-Hydroxyethyl)-1-piperazine ethanesulfonic acid) CDM for the handling of early embryos (CDM supplemented with 0.5 mM D-fructose, 2.5% BSA-FAF (Bovine Serum Albumin-Fatty Acid Free), and 22.5 mM NaCl) and vortexed for 1 min to remove cumulus cells. Groups of 50 presumptive embryos were randomly transferred and cultivated on CDM-1 (CDM with non-essential amino acids, 10 μM EDTA (Ethylenediaminetetra-acetic acid), 5% FAF-BSA, 0.5 mM D-fructose, 2.77 mM myo-inositol, and 0.1 mM taurine) for 56 h in a gas mixture consisting of 5% CO_2_, 5% O_2_, and 90% N_2_ at 38.5 °C. Eight-cell embryos were incubated and selected on HEPES CDM for the handling of late embryos (CDM supplemented with 2 mM fructose, 2.5% BSA, 1.47 mM essential amino acids, and 26.5 mM NaCl). Embryos were cultured for up to 7 days (day 0 = in vitro insemination), and groups of 30 embryos from each round of fertilization were harvested at the following developmental stages: two to four cells (30–44 hpi), eight cells (90 hpi), compacted morula (120 hpi), and blastocyst (168 hpi) [8,9]. Before mRNA analysis, the embryos were washed three times in phosphate-buffered saline (PBS). Embryos were washed twice in Dulbecco’s PBS, then frozen and stored at −80 °C in RNAlater (Ambion, Grand Island, NY, USA). Three relative quantification assays were performed.

### 2.4. Immunohistochemical Analysis

For the examination of *ashwin* protein localization, embryonic and adult bovine ovaries and testes were dissected in sections no larger than 2 cm^2^. The tissue samples were placed in 10% buffered formalin for 24 h and then embedded in paraffin. Sections of 3 μm thickness were obtained and mounted on adhesive-coated glass slides for hematoxylin and eosin staining. Follicles were classified using the morphological criteria of granulosa cell shape and the number of layers surrounding the follicle [10].

Immunohistochemical analysis was performed with a specific polyclonal antibody against the *ashwin* marker (C2orf49 (k-12), Sc-137342; Santa Cruz Biotechnology, Santa Cruz, CA, USA). Tissue sections were deparaffinized in two changes of xylene and dehydrated in descending concentrations of ethanol until water only. Antigen retrieval was performed with 0.001 M EDTA at 80 °C, and the slides were then treated with 0.2% Triton-X100 (Bio-Rad, Hercules, CA, USA). After blocking with 10% bovine fetal serum (BFS)/5% milk for 1 h at room temperature in a humidified chamber, the tissues were incubated with the primary antibody diluted 1:100 in 1% BFS/PBS at 4 °C overnight. Immunodetection was carried out using a standard immunohistochemistry staining method labeled streptavidin-biotin (LSAB) system (cat. K0690; Dako) according to the manufacturer’s instructions. Diaminobenzidine was used as a chromogen, and sections were counterstained with Harris hematoxylin. To establish a negative control, the primary antibody was replaced with PBS buffer. Images were acquired at 40× magnification using a digital camera coupled to an optical microscope (AxioStar Plus; Carl Zeiss).

### 2.5. Statistical Analysis

To enhance the understanding of the function of the *ashwin* gene, expression profiles were compared among development stages in fetal and adult tissues. A Dunnett test was performed to determine whether differences existed among the samples. Pairwise multiple comparisons were performed for all pairs of means with an alpha level of 0.05. The least significant difference method was used to identify significant differences among mean values obtained using the PRISM software package (GraphPad Software, Inc., La Jolla, CA, USA).

## 3. Results

### 3.1. Ashwin mRNA Expression in Fetal and Adult Tissues

The expression of *ashwin* was detected in all fetal and adult bovine tissues analyzed; expression in fetal and adult brain tissues served as a reference for demonstrating its ubiquity (Figure 1a,b). In fetal tissues, the *ashwin* gene showed significant down-regulation in the ovary (1.7-fold), spleen (1.7-fold), and lung (1.3-fold), with clear up-regulation in the kidney (2-fold) and no significant difference in the testis, heart, or muscle (Figure 1c). A different expression pattern was observed in adult tissues, with the testicle (1.3-fold), spleen (2-fold), lung (1.5-fold), and muscle (1.9-fold) displaying up-regulation in comparison with the ovary, heart, and kidney, in which no significant expression change was detected (Figure 1d).

Significant differences in *ashwin* gene expression profiles were found in the same tissues among developmental stages, with relative expression levels declining from fetal to adult stages in the testis, muscle, and kidney, and increasing in the ovary, spleen, and lung. No significant expression change was detected in the heart or brain (Figure 1d).

### 3.2. Expression of Ashwin mRNA in Early Embryonic Development

The *ashwin* gene showed clear down-regulation over the course of embryonic development, with particularly high expression in two-, four-, and eight-cell embryos and no significant change at the morula or blastocyst stage (Figure 2).

### 3.3. Localization of the Ashwin Protein in the Ovary and Testicle

Positive immunoreactivity was observed in the oocyte cytoplasm of primordial germinal follicles and primary follicles distributed throughout the embryonic and adult ovarian cortices. In addition, immunoreactivity was detected in the secondary and mature antral follicles of the adult ovary (Figure 3). No positive signal was observed in the negative control.

In the testes, the presence of the *ashwin* protein was strongly observed in the region of the seminiferous tubules, particularly in the internal regions where spermatogenesis occurs (Figure 4). In the fetal testicle, a strong signal was focused on the prospermatogonia, a cell group that is abundant in this primary stage of development (Figure 4a). In the adult testicle, the positive signal was focused in the spermatocyte region, and absent in the region of the spermatogonia (Figure 4b).

## 4. Discussion

In the present study, we found that *ashwin* is expressed ubiquitously in the bovine ovary, testicle, heart, spleen, lung, muscle, kidney, and brain, except in oocytes and early embryos, indicating that the expression of the gene is down-regulated during embryonic genome activation, which is similar to the pattern reported in maternal genes [11,12,13,14]. Although the molecular mechanisms in which *ashwin* participates have not been elucidated, studies in *X. laevis* have shown that the overexpression of *ashwin* alone or in combination with noggin induces changes in gene expression in the embryonic ectoderm (e.g., zic 3, NCAM and noggin overexpression), demonstrating the synergistic activity of the two proteins [1]. Noggin supplementation of cultures for oocyte maturation and in vitro bovine embryo development induces overexpression of the maternal expression genes Mater and HSP70 [15]. Both proteins are important in embryonic development, the first as part of the subcortical maternal complex (SCMC), which regulates multiple developmental events during the oocyte–embryo transition, including the processing of the male genome, maternal mRNA degradation, DNA methylation, and genome activation [16,17]. The second protein is a thermal shock protein that allows the embryo to respond to environmental and chemicals changes, such as thermotolerance and protection against reactive oxygen species (ROS) [18]. Our results invite speculation that *ashwin* expression affects the proper functioning of the SCMC, embryo viability, and oocyte competence through the regulation of these genes. Further studies are needed to evaluate this possibility.

The *ashwin* protein was present in fetal prospermatogonia and adult spermatocytes. A recently published histological analysis demonstrated that spermatogenesis in the domestic yak (*Bos grunnies*) is comparable to that in *Bos taurus* [19]. RNA sequencing of gene expression dynamics during the prospermatogonia-spermatogonia transition and spermatogenesis in this domestic species revealed a transcriptomic signature specific to each process. The total sequences of Wang et al. [19] show positive *ashwin* sequencing, supporting our observation that this gene is expressed during the prospermatogonia-spermatogonia transition and in the spermatogonial differentiation phase. However, the spatiotemporal differences in the distribution of the protein detected by immunohistochemistry in the present work support the hypothesis that *ashwin* in adult tissue is related more to spermatogenesis function, and that in fetal tissue it is related more to the development and maintenance of spermatogonia. NCAM, which is also synergistically overexpressed due to the actions of *ashwin* and noggin [1], is an adhesion protein detected in the epithelium of seminiferous tubules that maintains the interaction between prospermatogonia and Sertoli cells, a process that regulates maturation and development of the prospermatogonia and prospermatogonia-spermatogonia transition [20]. Thus, elucidation of the possible relationships of *ashwin* and NCAM with cell adhesion mechanisms in the testis is of interest.

## 5. Conclusions

The relative quantification by qPCR and immunodetection of *ashwin* in this work allowed the identification of the presence and variable expression of the *ashwin* gene during bovine tissue development. This study also revealed a potential role for *ashwin* in the embryonic development of gametes, as well as in early embryonic development, which suggests a biological role in bovine reproduction through synergistic action with noggin on the control of the expression of relevant proteins (e.g., MATER, HSP70, and NCAM). Thus, the identification of SNPs (Single-nucleotide polymorphism) in this gene, and their relationship with the activity of these proteins, may enable its use as a molecular marker in future work seeking to improve fertilization in cattle, embryonic quality, or oocyte competence.

## Figures and Tables

**Figure 1 animals-10-00276-f001:**
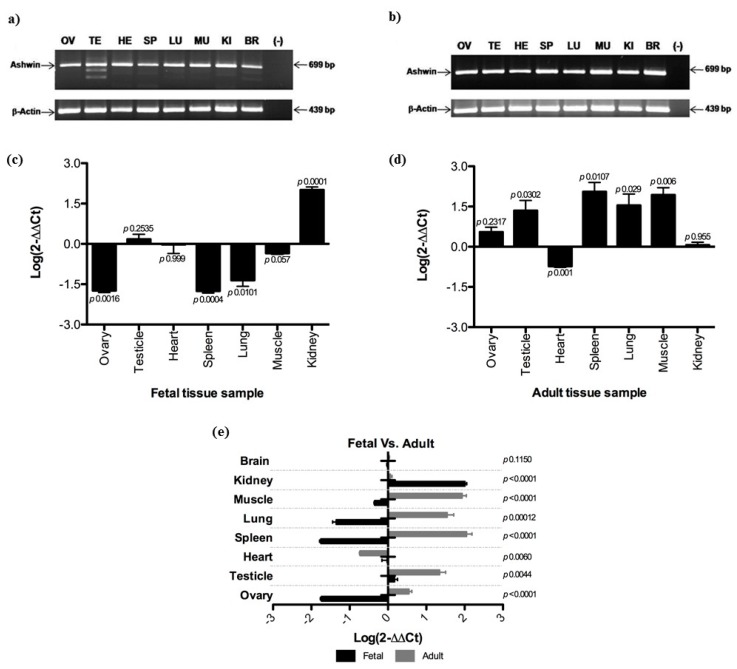
*Ashwin* expression profiles in fetal and adult tissues. (**a**,**b**) Agarose gel electrophoresis polymerase chain reaction (PCR) targeting genes encoding for *ashwin* and β-actin at the fetal and adult stages, respectively; (**c**,**d**) quantitative PCR results for *ashwin* gene expression in different tissues at the fetal and adult stages, respectively; (**e**) relative expression at the fetal and adult stages. Data were analyzed using a two-sample Student’s *t*-test. Data are presented as means ± standard deviations from two independent quantitative real-time PCR experiments (average of three independent reverse transcription reactions, each tested with three PCRs). Dunnett multiple comparison tests were performed, and expression in fetal and adult brain tissues served as a reference. *Ashwin* expression levels were normalized using the 2-ΔΔCT method.

**Figure 2 animals-10-00276-f002:**
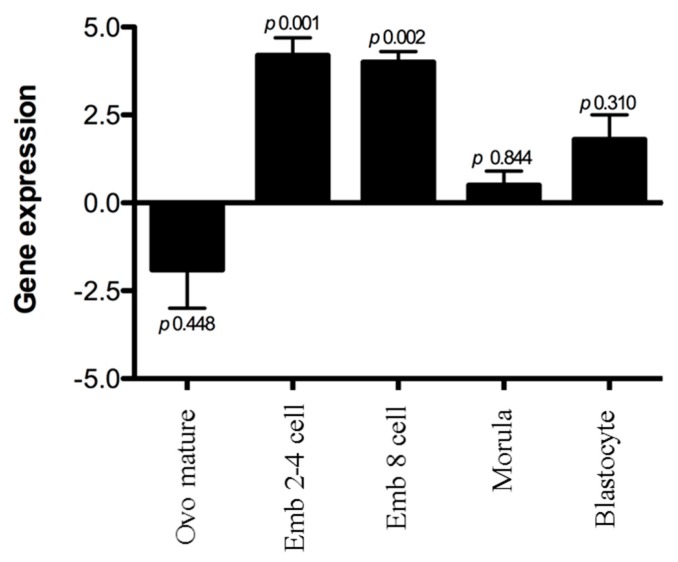
Expression of the *ashwin* gene in bovine embryos at different stages, normalized to expression levels of the *ashwin* gene using the 2-ΔΔCT method. Expression in immature oocytes served as a reference. Dunnett multiple comparison tests were performed.

**Figure 3 animals-10-00276-f003:**
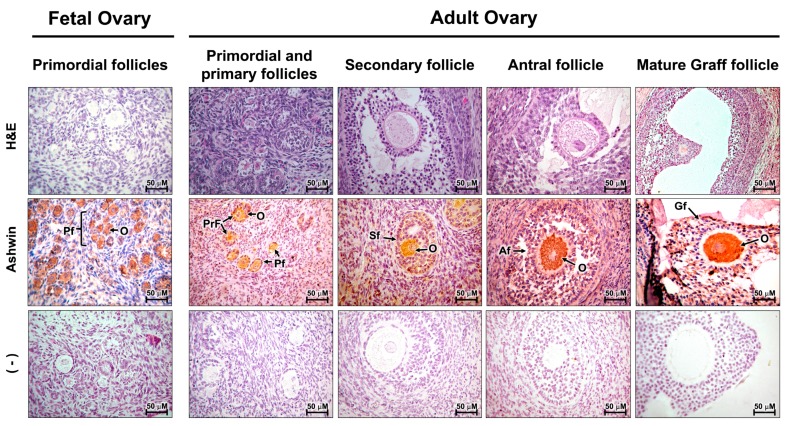
Expression of the *ashwin* protein in fetal and adult bovine ovary tissue. Representative immunohistochemical staining of bovine ovary sections showing *ashwin* immunoreactivity (brown) in the oocyte cytoplasm of primordial (Pf), primary (PrF), secondary (Sf), antral (Af), and Graff (Gf) follicles. (–) Negative control showing no positive staining. H&E, hematoxylin and eosin. The images were acquired at 40× magnification.

**Figure 4 animals-10-00276-f004:**
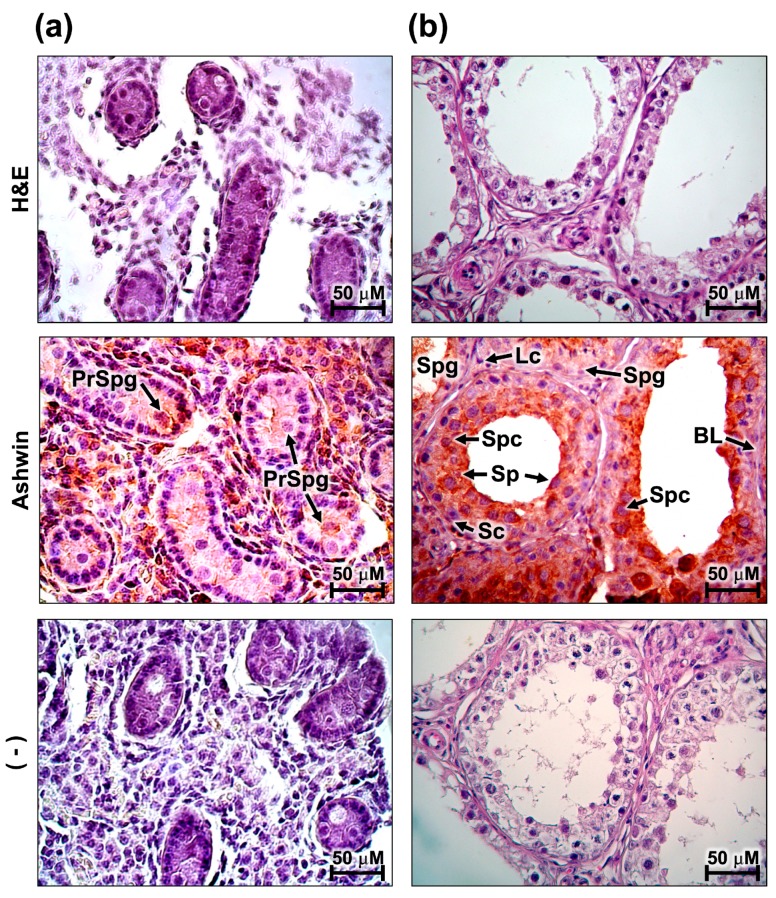
Representative images of immunostaining of the *ashwin* protein in bovine testis tissue. Positivity (brown) was observed inside the seminiferous tubules of fetal (**a**) and adult (**b**) testes. H&E, hematoxylin and eosin; PrSpg, prospermatogonia; Spg, spermatogonia; Lc, Leidy cells; Spc, spermatocyte; Sp, spermatid; BL, basal lamina; Sc, Sertoli cells. (–) Negative control. The images were acquired at 40× magnification.

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
