# Peer review of "Ashwin Gene Expression Profiles in Oocytes, Preimplantation Embryos, and Fetal and Adult Bovine Tissues"

_animals, 2020, doi:10.3390/ani10020276_

Round 1
Reviewer 1 Report
The manuscript entitled “Surveillance of ashwin gene expression profiles during bovine development” by Moreno-brito et al. describes the expression profiles of the Ashwin gene in oocytes, preimplantation embryos and in fetal and adult bovine tissues. The manuscript is well written, and provide the characterization of ashwin gene expression profiles through RT-qPCR and immunohistochemical analysis.
Major
The authors failed to provide the importance of the ashwin gene (general and for bovine specie). I would suggest to improve the contextualization of the Ashwin gene in the introduction with the knowledge of other species and highlighting the potential association of these findings with bovine specie. In addition, I strong encourage the authors improve the discussion. The authors failed in presenting the importance of the main findings of the work and what this knowledge could contribute to progress of science involved in the area.
Minor:
Title: Change for: “Ashwin gene expression profiles in oocytes, preimplantation embryos and fetal and adult bovine tissues”
Introduction
It is not clear the importance of the study of Ashwin gene in bovine specie. Improve introduction.
Material and methods
Provide an additional item for experimental design to favors readers understand the experiments. Sometimes it is not clear in which tissues or structures the analyzes were performed.
L74: replace the sentence for: “Samples of ovary, testis, heart, spleen, lung, muscle, kidney, and brain were collected from fetal (at 6thgestational month) and adult bovines at a local slaughterhouse.”
L77: Please, change the “polynerase” to “polymerase”.
L82: Authors described 100ng of RNA but in the line 94-95 a total of 250ng of RNA. Please double check.
L92-93: why RIN > 5.5?
L109-L110: Move this sentence to the immunohistochemical analysis description in order to improve description of the of the follicles characterization.
L110: provide number of total COCs used. How many IVF rounds were done? Provide description of oocyte collection and preparation for mRNA analysis (additionally provide number of oocytes used and replicates).
L117: how many bulls and which breeds?
L118: Please define SPF.
L120: improve description of the cumulus cell removal
L121: Please define the CDM and CD from Hepes CD.
L123: Define you day 0
L125: provide a reference of the in vitro production of bovine embryos (methodology) used by your group
L124: replace “to” by “cells,” and provide the moment specific the embryos were collected (in hours post-insemination - hpi)
L125-127: Do the authors remove the remaining cumulus cells (e.g. triapsin or pronase treatment)? Provide the number of embryos and replicates used for mRNA analysis.
L130: Immunohistochemical
L146: provide magnification used
L148-153: authors must improve the description. Which variable was evaluated by each statistical test? Why to different test and software?
Results: general comment: for the entire results session, please provide p values after in each comparison (i.e. greater, higher, up-or downregulation….). Additionally, specify the groups that you are comparing
L158: “down-regulation” compared with which tissue?
L159: “up-regulation” compared with which tissue?
L159: “no significant difference” compared with which tissue?
L161: testicle was not different from the ovary!
Figure 1c,d: I would suggest the authors to provide letters to indicate statistical differences. Or define a reference tissue expression (brain?) and compare all groups to the reference.
Figure 1e: please edit the image providing asterisk above the comparison that was statistically different.
L173: replace “different developmental stages” by “fetal and adult stages, respectively”
Figure 2: how authors have evaluated that the oocytes were matured to proceed the mRNA analysis? Why the mRNA abundance was reduced in matured oocytes?
L187: include “in oocyte cytoplasm”after “observed in”
Figure 3: provide scale bar in the image and magnification in the figure legend
L193: replace “in oocyte (O),” by “in oocyte cytoplasm of”
L203: delete “embryonic”
Discussion: general: The present reviewer strongly encourage the authors to improve the entire discussion session
L210-211: it is not clear that Ashwin is a maternal gene based only in the expression pattern similarity with Mater gene… Please improve the text and argument
L214-217: this is speculative. The authors did not evaluated this.
L237-238: how?
Author Response
We upload the document

Reviewer 2 Report
In this manuscript, Moreno-Brito et al., examined the expression pattern of Ashwin in different cattle tissues and found that this gene is expressed in preimplantation bovine embryos, testis and ovaries. This study has a very clear aim and the experiments were properly conducted. Several points should be addressed carefully before publication. Suggestions and concerns are listed below with no particular order:
Abstract:(2) “candidate maternal gene” is difficult to understand. Ashwin expression was decreased in mature oocytes then increased in 4 and 8 embryos. (3) “present …in meiosis…” lacks evidence. From the image provided in the manuscript, it is impossible to tell where the protein is localized in spermatocytes in bovine testis. (1) “functional relevance” is confusing because this paper only described expression of this gene in some bovine tissues. Materials and Methods How many animals were used? Which breed? Did the authors sequence the PCR product? Did the authors examined antibody specificity? Results Data presentation should provide more detail. For example, line 179-181, “high expression” how many folds? Figure 1a, TE shows multiple bands. Testicle or testis? Figure 2. 2-4 cell and 8 cell embryo data missing standard error bar. Figure 3: clarity of Gf follicle image needs to be improved. Figure 4 (a): difficult to tell if this tissue is from testis, looks like epididymis. Cannot see a single germ cells. Figure 4(b): difficult to see spermatocytes and spermatids due to poor fixation. And it is difficult to see which cells are stained positive.
Discussion and conclusion “gonocyte” or “prospermatogonia” may be a better word to describe germ cells in fetal gonad because undifferentiated spermatogonia are developed from gonocyte in neonatal testis. “revealed a potential role” must be revised. This data did not provide any data from functional experiments.
Author Response
We upload the document.

Round 2
Reviewer 2 Report
Most of my previous concerns and suggestions are addressed. The quality of this manuscript is greatly improved. However, I still believe Figure 4(a) shows staining of epididymis, not semiferous cord of testis. I can not see a single "spermatogonium" in the section. Please confirm.
Author Response
Thank you for your observations. We have revised Figure 4a and we have update and revised the text L-222, L229